# Diverse Impact of E-Cigarette Aerosols on Oxidative Stress and Inflammation in Lung Alveolar Epithelial Cells (A549)

**DOI:** 10.3390/ijms262210967

**Published:** 2025-11-12

**Authors:** Maciej Roslan, Katarzyna Milewska, Piotr Szoka, Kacper Warpechowski, Urszula Milkowska, Adam Holownia

**Affiliations:** Department of Pharmacology, Medical University of Bialystok, Mickiewicza 2c, 15-222 Bialystok, Poland

**Keywords:** e-cigarette smoke, cytotoxicity, DNA damage, inflammatory markers, oxidative stress, subclinical inflammation

## Abstract

This study investigated the pro-inflammatory and pro-oxidative effects of popular electronic cigarette aerosols (ECAs) compared with conventional cigarette smoke (CS) in the cultured human alveolar epithelial cell line (A549). Using cytotoxicity assays and four ECAs, substantial differences in biological impact were observed. CS exposure led to significant declines in cell viability and pronounced morphological changes, consistent with the presence of toxic combustion byproducts. Most ECAs caused negligible cytotoxicity except for the tobacco-flavoured variant, which demonstrated marked toxicity. DNA damage and altered cell cycle profiles were minor. Oxidative stress analysis revealed stable superoxide dismutase activity but notable glutathione depletion, especially with watermelon- and strawberry-flavoured ECAs, and unaltered mitochondrial transmembrane potential, indicating the importance of individual flavour additives in cellular antioxidant defence. Inflammatory markers, such as TNF-α, NF-κB, and IL-6, were differentially elevated across the CS and ECA groups, with IL-6 consistently increased, underscoring its role in regulating epithelial cells. Advanced double fluorescence analysis revealed increased cellular heterogeneity and inflammation, which was distinct for all ECA flavours. Overall, the findings demonstrate considerable heterogeneity in biological effects among ECA flavourings and propose a simple ECA biomonitoring model. The results emphasise the necessity for individualised toxicity assessments, especially regarding subclinical inflammation and potential long-term health outcomes.

## 1. Introduction

Over the past two decades, e-cigarettes, also known as electronic nicotine delivery systems (ENDSs), have become a popular alternative to traditional cigarettes. Advertised as less harmful than conventional cigarettes, these devices have gained popularity among young adults and people trying to reduce or quit smoking [1,2]. However, e-cigarette ECAs) contain several potentially toxic compounds, including propylene glycol, vegetable glycerin, nicotine, carbonyl aldehydes, heavy metals, as well as various flavouring compounds, which cause adverse effects on the respiratory system, particularly in pulmonology patients [3,4,5]. Clinical research has evidenced that ENDS use brings significant risks, extending beyond the effects of nicotine [6]. Experimental studies have shown that ECAs induce metabolic changes in lung alveolar epithelial cells, provoke oxidative stress, decrease mitochondrial activity, and decrease mitochondrial transmembrane potential [7,8]. Moreover, short exposure of lung alveolar epithelial cells to EC upregulated genes important in oxidative stress and carcinogen metabolism and downregulated genes related to cytokine and chemokine signalling [9]. This may lead to disruptions in cellular energy homeostasis, increased production of reactive oxygen species (ROS), disruption of the antioxidant balance and activation of oxidative stress [10,11]. Comparative studies indicate that, under conditions of acute exposure, e-cigarette aerosol has lower cytotoxicity than traditional cigarette smoke (CS), but it can still lead to DNA damage in the form of double-strand breaks and DNA adducts [12,13,14]. The observed effects of ECAs on A549 cells are largely dependent on the presence of nicotine [15]. An interesting toxicological aspect of e-liquids is the presence of flavouring substances like cinnamaldehyde, menthol, vanillin, and ethyl maltol. Recent studies with alveolar epithelial cells demonstrated significant cytotoxicity of various EC flavourings with notable differences between flavourings [16]. These compounds are safe when ingested but were not tested in inhalation exposure and demonstrated significant cytotoxic effects in cell cultures, including A549, by increasing oxidative stress and intensifying inflammatory processes [17,18,19]. Furthermore, some flavours induce epithelial–mesenchymal transition that promotes migratory and invasive cell features. Studies on the common effects of ECAs and tobacco smoke are few and often limited to single markers [20,21]. On the other hand, the involvement of smoking, ROS, and inflammation in cancer initiation is greatly recognised and widely discussed [22], but the precise role of chronic subclinical inflammation due to EC smoking is still not recognised.

This study aimed to assess the pro-inflammatory and pro-oxidative effects of popular ECAs in cultured human epithelial cells, compare those effects to classical tobacco smoke, point to the most hostile ECA, and define subsets of affected cells for a possible subclinical inflammation. By simultaneously analysing oxidative stress and IL-6 expression, this research seeks to identify the flavour variants with the highest biological impact and to propose new cellular endpoints for potential subclinical diagnostic use. These results provide essential insights into the differential safety profiles of ECAs and support the need for comprehensive toxicological scrutiny and bespoke public health guidelines.

## 2. Results

Our results are organised into four panels. Panel 1 (Figure 1 and Figure 2) shows data related to cell morphology, growth, cytotoxicity, and DNA damage; panel 2 contains oxidative stress-related parameters (Figure 3); panel 3 encloses inflammation-related parameters (Figure 4); and panel 4 (Figure 5 and Table 1) integrates data from the comparison of the effects of ECAs 1–4 on oxidative stress vs. IL-6 on binary scatterplots.

### 2.1. Panel 1: Cell Morphology, Growth, Cytotoxicity, and DNA Damage

#### 2.1.1. Morphology of A549 Cells

Figure 1 shows the morphology of control A549 cells (Figure 1A), cells exposed for 24 h to cigarette smoke (Figure 1B) and cells exposed for 24 h to ECA 1 (Figure 1C). Magnification 200X, Giemsa–Wright staining.

Control cells (Figure 1A) have typical lung epithelial cell morphology with a characteristic polygonal shape, clearly defined cell boundaries, and uniform distribution on the culture surface. Cell nuclei presented a normal structure with evenly distributed chromatin. Significant morphological changes are observed in cultures grown in CS-conditioned medium (Figure 1B). Reduced cell density, irregular shape, and the presence of numerous cells with degenerative features were observed. An increased number of cells detached from the bottom of petri dishes was also observed. Cells exposed to ECA 1 (Figure 1C) had reduced cell confluence and the presence of single, rounded cells. Similar, but less pronounced, changes were observed with ECA 4, while ECA 2 and ECA 3 showed the least deviation from the control morphology.

#### 2.1.2. Cell Viability (MTT Test)

Figure 2A shows cell viability by MTT assay after 24 h exposure to CS or ECAs 1–4. The MTT reduction assay was performed to reflect post-exposure cell numbers. The control cell values were expressed as 100% viability, and other results are presented as a percentage of the control. Exposure to CS significantly reduced cell viability by about 80% (*p* < 0.01), and ECA 1 decreased viable cell numbers by about 19% (*p* < 0.05). A Bonferroni post hoc test revealed no significant differences between the ECAs 1–4.

#### 2.1.3. Cytotoxicity/Analysis of Cell Cycle

Figure 2B–E shows the flow cytometry analysis of A549 cell cytotoxicity/proliferation. Cytotoxicity was assessed by flow cytometry analysis of histograms of propidium iodide-stained cells, determining the percentage of damaged cells (C) in the subdiploid G0/G1 phase (early G0/G1) as an indicator of cell damage, resting cells (G0/G1; D), and S-phase cells + G2/M cells (E). CS produced 42% toxicity (necrosis/apoptosis) (*p* < 0.01), and ECA 2 reached about 8% of DNA-damaged cells (*p* < 0.05). No toxicity was detected with other ECAs. CS and much less ECA 4 reduced the number of resting cells (*p* < 0.01 and *p* < 0.05, respectively).

#### 2.1.4. Cell Proliferation

Figure 2D,E show the number of resting cells (2D) and the fraction of proliferating cells. In the control group, the percentage of resting cells was more than 63%, and CS decreased the number of resting cells to approximately 48% (*p* < 0.01). A significant decrease (*p* < 0.05) was also observed with ECA 4 but not with other ECAs. Both CS and ECA 4 decreased cell proliferation by about 65% (*p* < 0.01) and 18% (*p* < 0.05), respectively. In contrast to CS, other ECAs again had no significant effect on proliferating cell fractions.

### 2.2. Panel 2: Oxidative Stress

#### 2.2.1. Intracellular Oxidative Stress

Figure 3A shows intracellular oxidative stress produced by 24 h CS or ECAs 1–4 exposure. Increased pro-oxidative effects were observed in each group. CS elevated DCF fluorescence approximately 3.5 times (*p* < 0.01), while ECA exposure resulted in smaller but highly significant values, reaching about 2–2.5 times higher (*p* < 0.01) fluorescence.

#### 2.2.2. SOD Activity

There was no significant difference between ECAs 1–4. SOD activity was not affected by CS nor by ECAs (Figure 3B).

#### 2.2.3. Intracellular GSH

Intracellular GSH levels significantly decreased (Figure 3C) in cells exposed to CS (by about 26%; *p* < 0.01) or ECA 3 and ECA 4 (both by about 17–18%; *p* < 0.05).

#### 2.2.4. Mitochondrial Transmembrane Potential

Exposure to cigarette smoke (CS) resulted in a marked depolarisation of mitochondrial membranes, whereas electronic cigarette aerosols (ECAs) induced much milder effects (Figure 3D). Notably, the reduction in the red/green fluorescence intensity ratio following CS exposure would be largely explained by a pronounced decrease in the number of J-aggregates, indicative of mitochondrial impairment. In comparison, ECA exposure failed to produce statistically significant changes in the fluorescence ratio, though it was associated with a modest reduction in red fluorescence, reflecting fewer J-aggregates. Furthermore, there were no significant differences in mitochondrial ΔΨM among samples exposed to various ECA formulations (ECAs 1–4).

### 2.3. Panel 3: Inflammation

#### 2.3.1. Intracellular IL-6

Figure 4A shows intracellular IL-6 in A549 cells exposed for 24 h to VCS or ECAs 1–4. IL-6 was measured in fixed and permeabilised cells using specific monoclonal antibodies and flow cytometry detection as described in Materials and Methods. IL-6 increased in all groups to a similar extent (by 45–74%; *p* < 0.01), and there was no difference between EC groups and CS.

#### 2.3.2. Intracellular TNF-α

TNF-α was elevated (Figure 4B) in cells exposed to CS (*p* < 0.01) and in cells treated with ECA 3 (*p* < 0.05). The results of ECA 1 (*p* < 0.05) and ECA 4 (*p* < 0.01) were significantly different from CS.

#### 2.3.3. Intracellular NF-κB

NF-κB was significantly increased (Figure 4C) in almost all groups to a similar extent (by 20–45%; *p* < 0.01) except for ECA 2, where the increase was not significant, probably due to higher standard deviation in this group. There was no difference between the median values of the ECAs 1–4 groups and CS.

### 2.4. Panel 4: Double Fluorescence Scatterplots

Figure 5 and Table 1 show representative double fluorescence scatter plots of A549 cells grown with ECAs 1–4 or CS-conditioned media for 24 h. Then, the cells were fixed, permeabilised, stained with a green fluorescent DCFDA for oxidative stress and with red peridinin chlorophyll protein-cyanine 5.5 (PC5.5)-bound monoclonal antibody for IL-6 (1:200; Cell Signalling, Danvers, MA, USA) and analysed with flow cytometry in scatterplot mode. Scatter area, vector size (distance between the central density point of control cells, Figure 5A, and the corresponding central density points of ECAs 1–4 exposed cells (Figure 5B)), and slopes of central tendency lines (symmetry line from central density point to the highest plot values) are shown in Table 1.

#### 2.4.1. Scatter Area

Only ECA 4 significantly increased cell scatter area by about 15%, and the area of cell scatter in the ECA 4 group was different from those in ECA 1 (*p* < 0.05) and ECA 2 (*p* < 0.05) but not ECA 3.

#### 2.4.2. Vector Size

Considering vector size (Figure 5B), the highest value was found in the ECA 3 group, significantly higher than ECA 1 (*p* < 0.01), ECA 2 (*p* < 0.01), and also higher than ECA 4 (*p* < 0.01).

#### 2.4.3. Slopes of the Central Tendency Line

There was no significant difference between the slopes of the central tendency lines.

#### 2.4.4. Cell Distribution

Considering cell distribution: in zone I (high inflammation, low oxidative stress), there were more cells after ECA 2 (*p* < 0.05) and ECA 4 (*p* < 0.01) exposure compared to ECA 1 or ECA 3. In zone II (very high inflammation, lower but high oxidative stress), ECA 2, ECA 3, and ECA 4 produced significant cell accumulation, with the highest increase compared to ECA 1 and ECA 4 (*p* < 0.01; by more than 100%). In zone III (low inflammation, low oxidative stress), there was an accumulation of cells treated with ECA 4 (*p* < 0.01; compared to all other ECA treatments). In zone IV (lower inflammation, higher oxidative stress), only ECA 4 significantly decreased (*p* < 0.05) cell numbers.

The II/IV ratios for highly fluorescent cells evidenced that ECA 4-exposed cells had relatively more inflammation than ECA 1 or ECA 2 (*p* < 0.01). Similar, but less expressed changes were noticed with ECA 3 (*p* < 0.05 for both ECA 1 and ECA 2).

## 3. Discussion

The association between smoking, ROS, and inflammation in cancer initiation is well established and extensively discussed in the scientific literature [23,24]; however, the potential role of chronic subclinical inflammation resulting from EC use remains inadequately characterised [25]. This study aimed to assess the pro-inflammatory and pro-oxidative effects of widely used EC aerosols in cultured human epithelial cells, compared with conventional CS, and to develop a diagnostic framework for identifying the most deleterious EC formulations and defining cell subsets for prospective targeting. Embedded within preclinical and in vitro research, our objective was to highlight new analytical aspects and introduce innovative tools applicable to routine investigation. We began with classical cytotoxicity tests using a human alveolar epithelial cell line, CS-conditioned medium and four EC aerosols. Our results showed significant differences in the cytotoxic effects of conventional CS and ECAs on A549 cells. The drastic decrease in cell viability after exposure to CS confirmed the strong cytotoxicity of tobacco smoke, well documented in the literature [26]. This effect can be attributed to numerous toxic compounds produced during tobacco combustion, including aldehydes, polycyclic aromatic hydrocarbons, and carbon monoxide [27]. Unlike traditional cigarettes, ECAs did not significantly affect A549 cells, except for ECA 1 (tobacco flavour), which reduced cell viability. This may be related to the specific chemical composition of tobacco-flavoured liquids containing compounds intended to mimic natural tobacco flavour. Published data on ECAs’ effects on A549 cells, including cytotoxicity and inflammation, are inconsistent [28,29]. Our results indicate significant cell damage after CS exposure. ECA 2 caused some DNA damage, but no toxicity was detected with the other ECAs. Regarding cell proliferation, CS and, to a lesser extent, ECA 4 reduced the number of proliferating and resting cells. It should be stressed that the proliferative nature of A549 cells affects the MTT assay. Microscopic observations confirmed the cytotoxicity results, showing distinct morphological changes in CS-exposed cells and more subtle changes in those exposed to ECA variants.

Cell cycle analysis revealed only slight changes in ECA-exposed groups and a significant alteration in CS-treated cells. This pattern is often observed in response to genotoxic stress and may indicate activation of DNA repair mechanisms or initiation of neoplastic processes if repair fails [30]. CS also decreased the number of resting cells, with ECA 4 also showing a significant decrease, unlike other ECAs.

Oxidative stress is implicated in cigarette smoke mechanisms and inflammatory lung diseases [31]. Pro-oxidants and inflammatory mediators such as TNF-α activate transcription factors like NF-κB and IL-6, leading to expression of pro-inflammatory genes and activation of pro-inflammatory cells [32]. IL-6 is a pro-inflammatory cytokine which induces the expression of a variety of proteins responsible for acute inflammation and plays a role in the proliferation and differentiation of cells [33]. TNF-α is a pleiotropic cytokine identified as a major regulator of inflammatory responses [34], while NF-κB is a key transcription regulator of immune responses, inflammation, and cancer [35].

ROS can also be generated intracellularly in mitochondria and via enzymatic systems such as xanthine/xanthine oxidase [36,37]. We have shown that CS is able to decrease mitochondrial ΔΨM. The role of ECAs is less evident, although some studies have shown that CEA can mimic CS [38]. Analyses of intracellular oxidative stress yielded unexpected results. Despite significant differences in cytotoxicity and proliferation, SOD activity remained unaffected and similar across experimental groups. This might suggest that tested substances do not directly generate reactive oxygen species or that A549 cells possess efficient antioxidant defence systems maintaining SOD homeostasis. Glutathione levels, particularly the significant reduction observed with watermelon- and strawberry-flavoured ECs, indicated a specific effect of flavouring ingredients on cellular thiol systems. Glutathione is crucial for antioxidant defence and xenobiotic detoxification, and its depletion may increase cellular sensitivity to oxidative damage [39].

Oxidative stress and inflammatory responses are tightly linked [40] and play a role in CS toxicity [41]. In our experiments, TNF-α was elevated in cells exposed to CS and those treated with ECA 3 and ECA 4. Results for ECA 1 and ECA 4 significantly differed from CS. Another pro-inflammatory molecule, NF-κB, significantly increased in almost all groups except ECA 2, where the increase was not significant. Only IL-6 increased similarly across all groups, with no difference between ECA groups and CS. IL-6 modulates epithelial cell proliferation, barrier function, and differentiation [42]. It promotes repair and proliferation after injury but can increase tight junction permeability during chronic inflammation, disrupting the epithelial barrier [43]. IL-6 also regulates proteins critical to epithelial cell structure [44]. The observed differences between EC flavour variants are important, as each exhibited a mildly different biological action likely related to their chemical compositions. ECA 1 showed the highest cytotoxicity, possibly due to tobacco-mimicking compounds. The watermelon flavour showed a unique profile; despite lacking significant cytotoxicity in the MTT assay, it caused noticeable morphological changes and increased cell proliferation, possibly due to fruit-derived flavouring effects on cellular metabolism. Menthol, despite lacking significant cytotoxicity, caused significant glutathione level changes, potentially linked to its chemical properties impacting membranes and intracellular transport. The literature suggests menthol-flavoured EC may affect membrane permeability and detox enzyme activities [45].

Oxidative stress and IL-6 were consistently elevated in A549 cells cultured with ECAs, so both were analysed via double fluorescence scatterplot experiments. Cell scatterplots between oxidative stress and inflammation show unique cell distribution with differing features, like scatter area and vector size. Only ECA 4 significantly increased the scatter area compared to ECAs 1 and 2. The highest vector size value was found in ECA 3, suggesting its utility in routine ECA analyses. Cell distribution analysis highlighted zones II and IV as most important (higher inflammation/oxidative stress ratio and lower inflammation to oxidative stress, respectively). ECAs 2, 3, and 4 produced significant cell accumulations, with ECA 4 showing the highest increase. In zone IV, only ECA 4 significantly decreased cell numbers. The distinct characteristics of strawberry-flavoured cigarettes (ECA 4) have likewise been reported by other studies [25,46,47]. The II/IV ratio for highly fluorescent cells indicated ECA 4-exposed cells had stronger inflammation than oxidative stress, more than ECAs 1 or 2. Similar but less pronounced changes occurred with ECA 3.

### Study Limitations and Future Directions

This study has limitations. Primarily, the use of a single cell line (A549) may not capture the diversity of lung cell responses. Future work should include broader cell panels and primary lung epithelial cultures. Also, exposure time was limited to 24 h, potentially insufficient to simulate long-term effects typical of real-world EC and cigarette use. Long-term studies should investigate cumulative effects and cellular adaptation mechanisms. Oxidative stress provides a strong signal compared to the weaker “molecular” signal of IL-6. More biomarkers could be considered. Interpreting in vitro effects, particularly translating subclinical to clinical meaning, remains challenging; however, future studies should explore broader biomarker standardisation.

Our findings have public health implications for EC regulation. Although ECs showed less direct cytotoxicity than conventional cigarettes, their effects on proliferation and potential epigenetic changes may pose long-term health risks. Different flavour ingredients show varying toxicity profiles, complicating safety assessments [45]. There is a need for caution in promoting ECs as safer alternatives, and regulation should address both direct toxicity and proliferative/epigenetic effects.

## 4. Materials and Methods

### 4.1. Reagents

All chemicals used in this study were obtained from Sigma Chemical (Poznan, Poland) unless otherwise noted, while culture media and cell culture reagents were purchased from GIBCO (Thermo Fisher Scientific, Waltham, MA, USA). Fluorescent antibodies were from Abcam (Cambridgeshire, UK), Cell Signalling (Danvers, MA, USA), or SantaCruz (Dallas, TX, USA).

### 4.2. Cell Culture

A549 cells, obtained from the American Type Culture Collection (ATCC; Manassas, VA, USA), were maintained in Ham’s F-12K Nutrient Mixture (Sigma Chem. Co., Poznan, Poland) supplemented with 10% foetal bovine serum, 100 U/mL penicillin, 100 μg/mL streptomycin, and 2 mM L-glutamine (GIBCO/BRL; Grand Island, NY, USA). Cultures were performed in Falcon flasks (Fisher, Poznan, Poland) at 37 °C in an atmosphere containing 95% air and 5% CO_2_. Cells were grown as monolayers and were regularly trypsinised and replated before reaching full confluence. After the third passage, cells were allowed to adhere, then maintained overnight in serum-free medium, and then subjected to further procedures. In some experiments, A549 cells were seeded into 6-, 24-, or 96-well plates and cultured for 24 h in control medium, cigarette smoke-conditioned medium, or e-cigarette aerosol-conditioned medium (flavours: 1-classic tobacco, 2-menthol, 3-watermelon, 4-strawberry) according to the procedure described in Section 2.3. All experiments were conducted under serum-free conditions for 24 h after replacing the standard culture medium with the appropriate conditioned medium.

### 4.3. Preparation of ECAs and CS-Conditioned Media

The ECA-conditioned media were prepared based on previously established procedures [48]. The aerosol was generated from P1 brand liquids with a nicotine concentration of 12 mg/mL in four flavours, 1–4, as described in Section 2.2. These flavours were selected as representative of the various flavour categories and compositions of e-cigarette liquids. The aerosol was passed through 100 mL of the culture medium using a low-pressure vacuum pump, maintaining parameters that allowed for comparable aerosol generation times to those used for traditional cigarettes. CS smoke-conditioned medium was prepared using two Marlboro Red cigarettes (Philip Morris, Cracow, Poland; full strength, with filters removed). Smoke was passed through 100 mL of culture medium using a low-pressure vacuum pump set to approximately 1 min per cigarette and sterilised by filtration using 0.22 μm filters.

### 4.4. Cell Exposure

All conditioned media were sterilised using 0.22 μm filters and immediately used for cell culture experiments. In each experiment, cells were seeded into 6-, 24-, or 96-well plates and cultured for 24 h in control or conditioned media.

### 4.5. Microscopy and Cell Morphology

After 24 h of incubation in the appropriate conditioned media, Giemsa–Wright staining was performed. Cells were observed under a light microscope, assessing both morphology and cell growth under each experimental condition.

### 4.6. Cell Viability Test

Cell viability was assessed using a mitochondrial activity-dependent reduction of MTT (3-(4,5-dimethylthiazol-2-yl)-2,5-diphenyltetrazolium bromide) to violet formazan. The assay was performed according to standard procedures after a 24-h incubation [49].

### 4.7. Flow Cytometry—Cell Cycle Analysis

Cell proliferation and cytotoxicity were assessed by flow cytometry using propidium iodide DNA staining and cell cycle analysis [50]. Representative flow cytometric histograms of propidium iodide fluorescence distributions are presented. Cells were classified based on their relative distribution in the individual cell cycle phases: damaged, subdiploid G0/G1 zone (early G0/G1 cells)—cytotoxicity/apoptosis, diploid zone (G0/G1—before DNA synthesis/resting), S phase (DNA synthesis)—proliferation/cell cycle arrest, and G2/M phase (after DNA synthesis/mitosis)—proliferation/cell cycle arrest. Each histogram was generated from the analysis of 5000 cells, with six samples analysed in each group. Proliferation was defined as the sum of cells in S + G2/M phases, while cytotoxicity was defined as cells in the early G0/G1 phase. Positive controls for DNA damage were obtained with 2.5 μg/mL cisplatin.

### 4.8. Assessment of Oxidative Stress

Intracellular oxidative stress was assessed using fluorescent DCFDA (2′,7′-dichlorodihydrofluorescein diacetate) staining [51]. DCF fluorescence histograms reflected the level of oxidative stress. Increased oxidative stress was characterised by a rightward shift in the fluorescence peak and mean fluorescence intensity. Positive controls were obtained with 50 μM tert-butyl hydroperoxide.

### 4.9. Glutathione Level

Intracellular glutathione levels were determined using fluorescent glutathione-specific antibodies (Abcam, Glutathione Assay Kit, Fluorometric, Cambridgeshire, UK). Measurements were performed by Beckman Coulter CytoFlex flow cytometer (Beckman Coulter, Warsaw, Poland) and Kaluza Software (version 2.1.2). 

### 4.10. Superoxide Dismutase Activity

Superoxide dismutase (SOD) activity was assessed using the Superoxide Dismutase Activity Assay Kit (Sigma-Aldrich CS0009, Merck Life Science Sp., Poznan, Poland) according to the manufacturer’s instructions.

### 4.11. Mitochondrial Transmembrane Potential

Mitochondrial transmembrane potential (ΔΨ_M_) was assayed in A549 cells using a red/green fluorescent lipophilic, cationic dye—5,5,6,6′-tetrachloro-1,1′,3,3′ tetraethylbenzimi-dazoylcarbocyanine iodide (JC-1) [38]. The red/green fluorescence ratio in the mitochondria reflected mitochondrial polarisation/depolarisation, respectively. Cells were stained with 2 μM JC-1, incubated for 20 min, washed with PBS and analysed with the Beckman Coulter CytoFlex flow cytometer (Beckman Coulter, Warsaw, Poland) on FL1 (green) and FL2 (red) channels. Data are shown as a red/green fluorescence ratio.

### 4.12. Intracellular IL-6, TNF-α, and NF-κB 

Intracellular IL-6, TNF-α, and NF-κB levels were measured to evaluate ECA-specific pro-inflammatory immune responses. After culturing, the cells were fixed for 10 min in 4% methanol-free formaldehyde at room temperature and stained by direct labelling with anti-IL-6 monoclonal antibody (Cell Signalling, Danvers, MA, USA), TNF-α (Abcam, Cambridgeshire, UK), or NF-κB (Santa Cruz, Dallas, TX, USA) antibodies bound to different fluorochromes. Antibodies were diluted (1:200, 1:500, and 1:100 for IL-6, TNF-α, and NF-κB, respectively) in permeabilisation/wash buffer with 0.1% Triton-X-100 (Sigma Chemical Company, Poznan, Poland) with 1% bovine serum albumin. Samples were incubated for 20 min at room temperature in the dark, washed, filtered via a 100 μm cell strainer (Fisher Scientific, Poznan, Poland) to remove aggregates, and 5.000–10.000 cells were analysed by a Beckman Coulter CytoFlex flow cytometer (Beckman Coulter, Warsaw, Poland). Doublet discrimination was manually performed by plotting FSC signal area versus FSC signal height on a linear scale. Positive controls were obtained with (100 ng/mL) lipopolysaccharide (LPS; from *E coli*. O111:B4, Sigma-Aldrich L2630, Poznan, Poland) applied to the cells for 24 h. Evaluation of IL-6, TNF-α, and NF-κB was based on right shifts of fluorescence intensity histograms versus the negative control.

### 4.13. Binary Fluorescence Scatterplots of IL-6 vs. Oxidative Stress

Changes in IL-6 expression and oxidative stress were assessed in double fluorescence experiments to delineate the relationship between both parameters and assess the response of particular cells. We addressed the issue by single and double staining the cells using DCFDA and epitope-specific rabbit monoclonal antibody (1:200) conjugated to peridinin chlorophyll protein-cyanine5.5 (PC5.5), from Cell Signalling (Danvers, MA, USA), as described earlier. Samples were analysed in flow cytometry (Beckman Coulter CytoFlex flow cytometer, Warsaw, Poland) on green and red channels. Experimental data were plotted as histograms, bivariate cytograms, and fluorescence density plots and were analysed for the spread, central tendency, and vectors using the FlowJo (V10.8.1, Ashland, OR, USA) and Flowing Software (v.2.5.1. freeware, Turku University, Finland).

### 4.14. Statistical Analysis

Statistical comparisons between groups were made using the one-sample Student’s t test, while multi-group comparisons were made using a one-way analysis of variance (ANOVA) with Bonferroni’s multiple comparisons. Differences between groups were considered statistically significant at a significance level of *p* < 0.05. All tests were performed as two-sided.

## 5. Conclusions

To the best of our knowledge, this is the first study to reveal significant differences in the biological effects of “watermelon-flavoured” versus “strawberry-flavoured” ECs. Unlike conventional cigarettes, which are known to cause acute toxicity, ECs appear to induce more subtle forms of inflammation and oxidative stress. These changes may lead to biochemical alterations that negatively impact consumer health, potentially contributing to subclinical inflammatory states that are not routinely detected by current clinical laboratory methods.

To address this gap, we introduced the concept of “diagnostic algorithms” designed to monitor these nuanced biochemical shifts. This approach is particularly relevant given that EC-related biomarkers are not yet part of standard diagnostic protocols. We have demonstrated significant biological differences between conventional CS and EC extracts on A549 cells, highlighting EC heterogeneity. Conventional cigarettes caused much greater direct cytotoxicity, but both CS and most EC variants affected proliferation, relevant to long-term cancer risk. EC flavouring ingredients exhibited diverse biological profiles, with tobacco flavour showing the highest cytotoxicity and fruit flavours (watermelon, strawberry) showing primarily proliferative effects. Finally, this study provides tools for biomonitoring diverse EC effects and emphasises individualised assessment of pro-oxidative and pro-inflammatory effects.

## Figures and Tables

**Figure 1 ijms-26-10967-f001:**
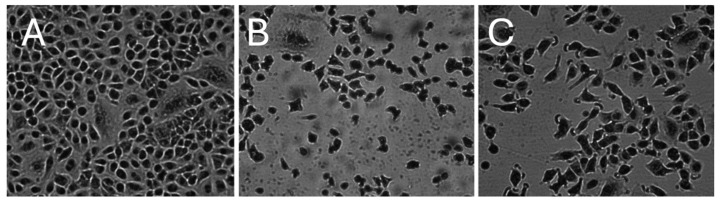
Morphology of control A549 cells (**A**), cells exposed for 24 h to cigarette smoke (**B**) and cells exposed for 24 h to ECA 1 (**C**). Magnification 200X, Giemsa–Wright staining.

**Figure 2 ijms-26-10967-f002:**
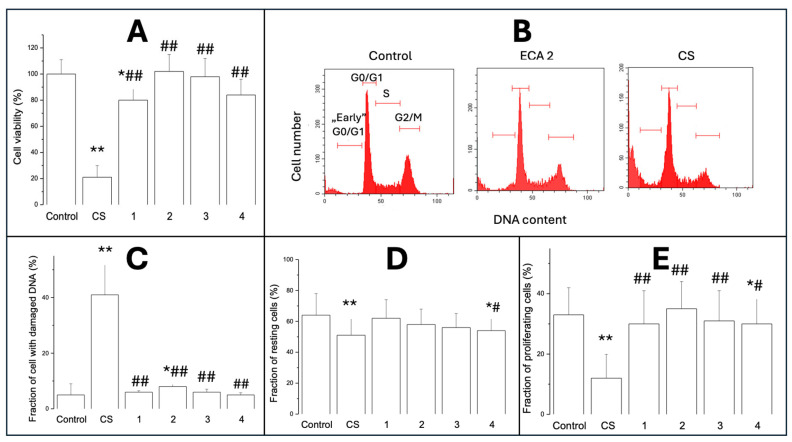
Cell viability—(**A**) (MTT test), DNA histograms of control cells, ECA 2, and CS exposed cells (**B**) (propidium iodide staining), with low molecular, damaged “early” G0/G1 fraction (**C**), resting cells (**D**), and proliferating cells at S and G2/M phases (**E**). Data are means ± SD of 5–6 assays. * *p* < 0.05; ** *p* < 0.01 for comparisons with the corresponding control cells; # *p* < 0.05; ## *p* < 0.01 for comparisons with the CS-treated cells.

**Figure 3 ijms-26-10967-f003:**
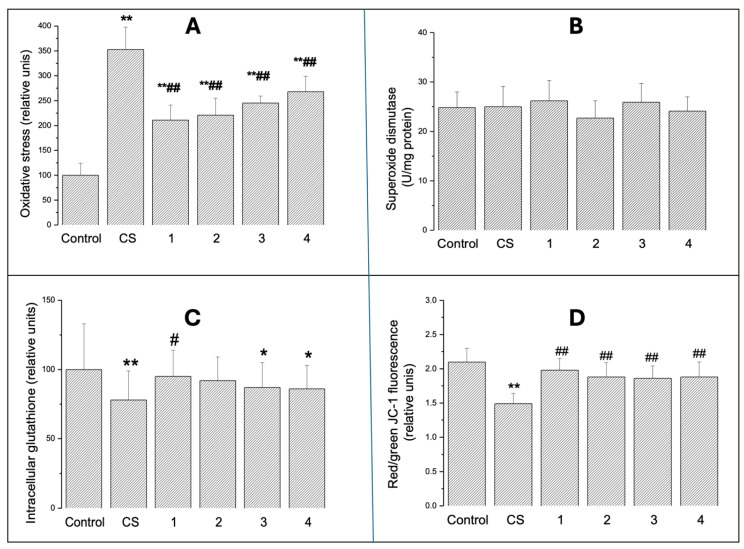
Oxidative stress in A549 cells exposed for 24 h to CS or ECAs 1–4 as described in Materials and Methods. Panel (**A**) represents DCF fluorescence, panel (**B**) is specific SOD activity, panel (**C**) shows intracellular GSH content, and panel (**D**) represents changes in mitochondrial transmembrane potential. Data are means ± SD of 5–6 assays. * *p* < 0.05; ** *p* < 0.01 for comparisons with the corresponding control cells; # *p* < 0.05; ## *p* < 0.01 for comparisons with the CS-treated cells.

**Figure 4 ijms-26-10967-f004:**
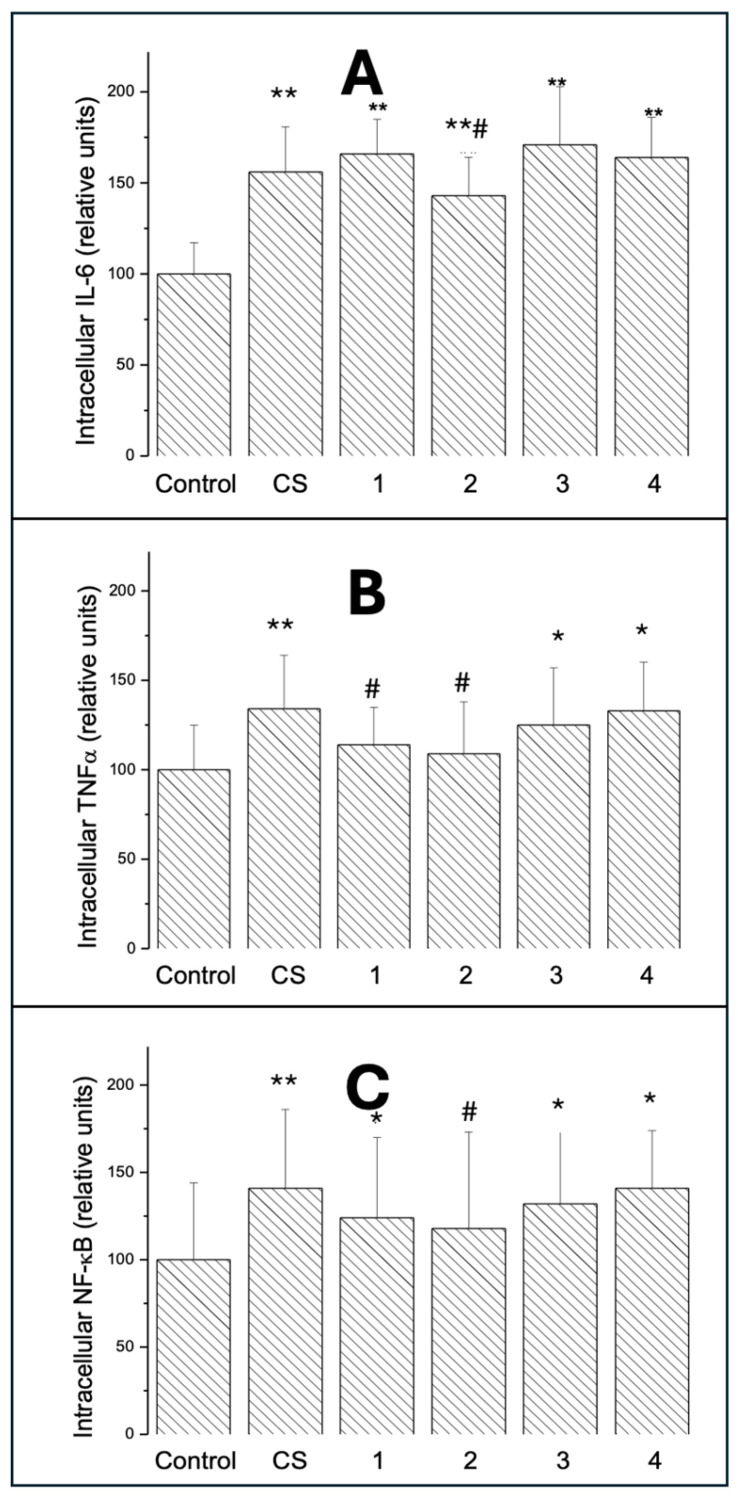
Intracellular IL-6 (**A**), TNF-α (**B**), and NF-κB (**C**) in A549 cells exposed for 24 h to CS and ECAs 1–4. All epitopes were measured in fixed and permeabilised cells using specific monoclonal antibodies and flow cytometry detection as described in Materials and Methods. Data are means ± SD of 5–6 assays. * *p* < 0.05; ** *p* < 0.01 for comparisons with the corresponding control cells; # *p* < 0.05 for comparisons with the CS-exposed cells.

**Figure 5 ijms-26-10967-f005:**
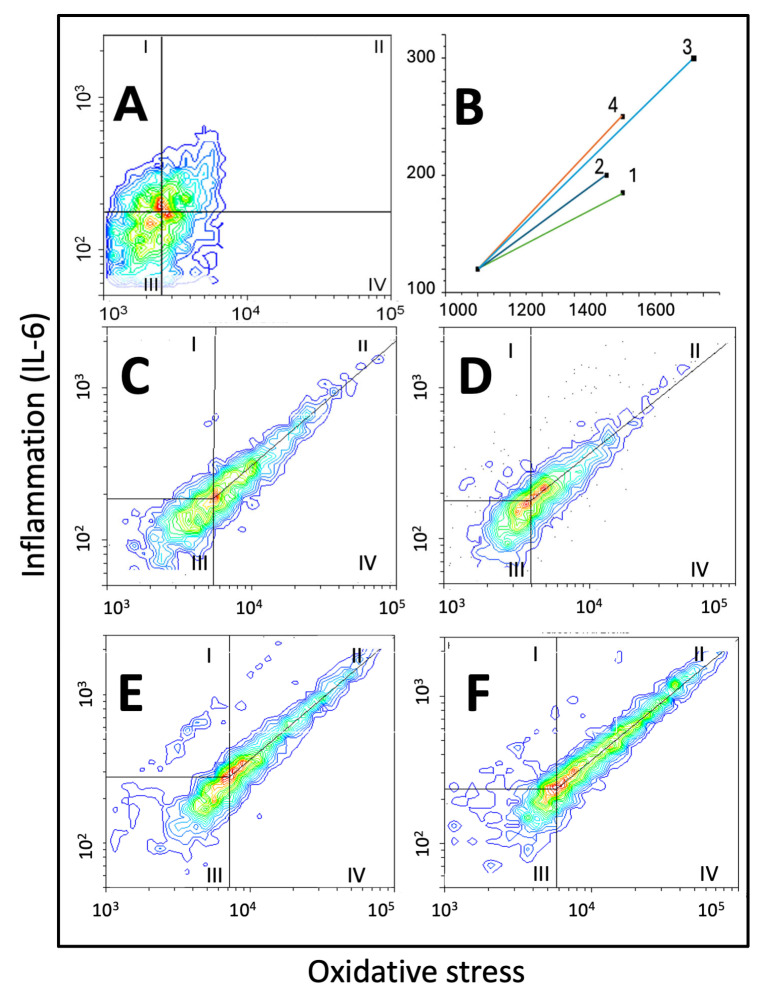
Representative double fluorescence scatter plots of A549 cells grown with ECAs 1–4 conditioned media for 24 h. The cells were fixed, permeabilised, and stained with a green fluorescent DCFDA for oxidative stress and with red peridinin chlorophyll protein-cyanine 5.5 (PC5.5) monoclonal antibody for IL-6 and analysed with flow cytometry in scatterplot mode. Scatter area, vector size (distance between the central density point of control cells (**A**) and the corresponding central density points of cells exposed to ECAs 1–4 (**B**)), and slopes of central tendency lines (symmetry line from central density point to the highest plot values (**C**–**F**)) are shown on panels (**C**–**F**) for ECAs 1–4, respectively.

**Table 1 ijms-26-10967-t001:** Data from analysis of double fluorescence scatter plots of A549 cells grown with ECAs 1–4 or CS-conditioned media for 24 h and double-stained with a green fluorescent DCFDA for oxidative stress and with red PC5.5 for IL-6 and analysed with flow cytometry in scatterplot mode.

Scatter Area, Vector Size, Slope of the Central Tendency Line, and Distribution of A549 Cells on Scatterplots of Oxidative Stress vs. IL-6				
(Relative Units)	ECA 1	ECA 2	ECA 3	ECA 4
Scatter area	101.1 ± 13.7	100.0 ± 9.1	110.2 ± 16.4	115.4 ± 10.2 *^
Vector size	1.0 ± 0.21	1.0 ± 0.19	1.83 ± 0.22 **^^	1.29 ± 0.17 *^^##
Slope of the central tendency line	0.90 ± 0.10	0.83 ± 0.11	0.97 ± 0.12	0.90 ± 0.15
Zone I cells	2.9 ± 0.31	3.7 ± 0.35 *	3.5 ± 0.40	4.0 ± 0.51 **
Zone II cells	8.9 ± 0.43	10.3 ± 0.97 **	12.6 ± 0.77 **^^	17.7 ± 0.61 **^^##
Zone III cell	11.2 ± 1.04	12.5 ± 0.98	12.3 ± 0.87	15.4 ± 1.45 **^^##
Zone IV cells	77.0 ± 9.4	73.5 ± 10.2	71.6 ± 9.1	62.9 ± 9.7 *
II/IV ratio	0.11 ± 0.05	0.14 ± 0.04	0.18 ± 0.03 *^	0.28 ± 0.06 **^^#

* *p* < 0.05; ** *p* < 0.01 for comparison with the ECA 1; ^ *p* < 0.05; ^^ *p* < 0.01 for comparison with the ECA 2; # *p* < 0.05; ## *p* < 0.01 for comparison with the ECA 3.

## Data Availability

The original data presented in this study are included in the article. Further inquiries can be directed to the corresponding author.

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
