# Peer review of "Diverse Impact of E-Cigarette Aerosols on Oxidative Stress and Inflammation in Lung Alveolar Epithelial Cells (A549)"

_ijms, 2025, doi:10.3390/ijms262210967_

Round 1
Reviewer 1 Report
Comments and Suggestions for Authors
Impact of e-cigarette aerosols on oxidative stress and inflammation in lung alveolar epithelial cells (A549): The role of subclinical inflammation
In this manuscript, Roslan et al, investigated the role of cigarette and e-cigarette aerosols on the viability, oxidative stress and cytokine production in A549 cells. The manuscript is very descriptive and does not delve into the mechanisms behind e-cigarette aerosol induced oxidative stress and cytokine production. The authors also use un-conventional methods for assessing oxidative stress and cytokine production.
Major concerns:
- Representative histograms for propidium iodide staining should be shown for control, cigarette and e-cigarette aerosol treated cells. There is no description of what the two panels in figure 2B represent.
- The significance of measuring superoxide dismutase and glutathione (GSH) as markers of oxidative stress is not elaborated. The authors need to explain the hypothesis and the reasoning behind using these markers as measurements of oxidative stress.
- What is the impact of e-cigarette aerosols on mitochondrial ROS? MitoSOX measurements would be helpful.
- The assay for measuring GSH is unconventional. Is the assay measuring reduced or oxidized GSH? A decrease in the ratio of reduced to oxidized GSH is usually a measure of oxidative stress. Simply a measure of GSH is not a measure of oxidative stress.
- The assay for measuring cytokines is also unconventional. What is the need to measure intracellular cytokine levels? Cytokines are secreted outside the cells to perform their functions, and the levels are conventionally measured by ELISA. I recommend measuring cytokines in the supernatant by ELISA. To supplement ELISA, qRT-PCR levels of the cytokines can be presented.
- The measurement of NF-κB is again unconventional? What are the authors detecting for NF-κB? P65 or p50? Are they measuring the phosphorylation status or the total protein? The measurement of NF-κB pathway should be assessed by western blotting for phospho-p65 and compared with the total levels of p65.
Minor corrections:
- E-cigarette aerosol abbreviations should be consistent (EC or ECA).
- Some words are unnecessarily capitalized (Example: lines 73-76). This should be corrected throughout the manuscript.
- The alpha (α) and kappa (κ) symbols for TNFα and NF-κB need to corrected (lines 274 to 278).

Author Response
We acknowledge the reviewers’ constructive feedback, which has significantly contributed to the improvement of the manuscript. In response, comprehensive revisions have been implemented. The updated version now includes additional experimental data, and the figures have been reformatted to enhance the clarity and focus on the principal objectives of the study, in accordance with the reviewers’ recommendations. Concerning major points:
- Representative histograms depicting propidium iodide staining have been incorporated, demonstrating the comparative effects on control cells, e-cigarette aerosol-exposed cells, and cigarette smoke-exposed cells. Detailed descriptions accompany these figures for comprehensive interpretation of the experimental outcomes.
- The revised Discussion chapter provides an expanded analysis of the utilisation of superoxide dismutase and glutathione as established biomarkers of oxidative stress, with appropriate citations to the relevant literature integrated to substantiate this role.
- To further elucidate mitochondrial involvement in redox imbalance induced by e-cigarette aerosols, supplementary experiments were conducted employing the JC1 mitochondrial probe. This assay discriminates green and red fluorescence to serve as an indicator of high and low mitochondrial transmembrane potential, thus enabling direct assessment of mitochondrial function. The associated data reinforce the mechanistic relevance of mitochondrial activity, and are discussed with reference to current literature. It should be noted, however, that a probe specific for mitochondrial superoxide dismutase was not available during the review period, representing a limitation of the present study.
- Glutathione Assay: A commercially available assay kit was employed for the quantification of total intracellular glutathione (GSH). This assay utilises monochlorobimane, a dye that forms a fluorescent adduct with glutathione via a glutathione-S-transferase-catalysed reaction. While the assay is straightforward to execute, it does not evaluate the GSH/GSSG (reduced/oxidised glutathione) ratio, which is a more informative marker for oxidative stress and mechanistic insight but requires an ELISA-based approach. The reduced form of glutathione (GSH) is recognised as the most prevalent intracellular antioxidant, where a decrease is widely accepted as an indicator of oxidative stress. Measuring the total GSH content was chosen because it serves as a universal parameter suitable for a double fluorescence flow cytometry (FC) model that allows detection of subtle differences induced by various electronic cigarette aerosols (ECA)
- Cytokine Assessment- The underlying principle for cytokine analysis is analogous. Since A549 cells are non-immune epithelial cells, inflammatory mediators are present but not abundant. Measurement of secreted cytokines was deemed inappropriate, as the focus is on intracellular changes rather than the extracellular milieu. While the activation status of cytokines is relevant, it is not universally applicable as an indicator. Additional explanatory notes regarding this topic have been appended to the Discussion section of the manuscript.
- NF-κB and Inflammatory Response NF-κB is a central transcription factor overseeing inflammation by regulating genes implicated in immune responses. It was hypothesised that NF-κB would mediate inflammatory signalling in response to all electronic cigarette exposures (ECEs) employed in this study. Contrary to this expectation, IL6 demonstrated consistent responsiveness to all ECE treatments, whereas NF-κB did not. Additional commentary concerning the role of IL6 in the inflammatory cascade has been incorporated into the Discussion. Please note that this work focuses more on new and sensitive diagnostics of ECA-induced early inflammation/red-ox imbalance than on the role of a particular mechanism.
Editorial and Minor Corrections
- Consistency in the abbreviations for e-cigarette aerosols has been established throughout the manuscript.
- Erroneous capitalisation resulting from PDF transformation has been rectified.
- The Greek letters alpha (α) and kappa (κ) are now properly used for TNFα and NF-κB throughout the text.
Additional spelling and grammatical corrections have also been implemented.
Reviewer 2 Report
Comments and Suggestions for Authors
Please describe the novelty of the study in the Introduction and Disscussion section as the influence of E-cigarettes on human lung epithelial cells was examined before as well as the cytotoxicity of vaping flavourings (https://www.sciencedirect.com/science/article/pii/S2214750025002100). The impact of long-term use is also described before (https://pmc.ncbi.nlm.nih.gov/articles/PMC9559034/). From the presented manuscript, there is no novelty in this research, thus the paper should be rewritten and resubmitted.
Also, the methodology for the “prospective diagnostic algorithm” is not explained and described. Explain this approach for "Cell polarisation between oxidative stress and inflammation". Has this approach been used before or this is new? Further, this cannot be regarded as algorithm, as this is only double fluorescence measurement. Based solely on in vitro results, is cannot be stated as diagnostic algorithm, these results need to be validated in vivo first and incorporated in the “risk algorithm”. Thus, the title of the paper is also not suitable and "The role of subclinical inflammation" should be removed, as this is only in vitro study.
Author Response
Thank you for your thoughtful and valuable comments on our manuscript. We appreciate your suggestions and have carefully addressed each of your points in the revised version.
- We confirm that the double fluorescence measurement offers sufficient sensitivity to differentiate among various ECA effects. The novelty and significance of our study have been further emphasised in the Introduction and Discussion sections, and appropriate references have been added to support these changes.
- Both articles you mentioned have been incorporated and appropriately cited in the Introduction and Discussion sections.
- The term “prospective diagnostic algorithm” has been removed to avoid overstatement. Instead, we now highlight the diagnostic implications of the observable changes that can be illustrated using fluorescence scatterplots in toxicological analyses.
- The phrase “cell polarisation between oxidative stress and inflammation” has been replaced with a more precise description that reflects the direct relationship between these parameters across different cell subpopulations.
- Regarding the issue of “subclinical inflammation,” we agree that in vitro studies should pave the way for research in the challenging area of moderate toxicity and inflammation. While the contribution of these factors to the development of subclinical inflammation warrants further investigation, we believe our model represents a valuable step forward in the field; however, we have removed “subclinical inflammation” from the title.
Round 2
Reviewer 1 Report
Comments and Suggestions for Authors
1) qRT-PCR should be performed for measuring IL-6 and TNFα levels. While ELISA or luminex is the appropriate method to assess cytokines, qRT-PCR is also widely accepted and a relatively easy and inexpensive method. I continue to maintain that qRT-PCR experiment should be performed as a further validation of an increase in inflammatory response.
2) In the absence of any information about the antibody used for NF-κB measurement it is impossible to conclude about its impact on inflammation. Total cellular levels of NF-κB has no relevance on inflammation. It is only when the NF-κB pathway is activated does it serve as a maker of inflammatory signaling. Activation of the NF-κB pathway is typically measured by assessing the phosphorylation status of p65 and p50 proteins. Phospho-p65 and/or p50 levels should be measured.
Author Response
We gratefully acknowledge the Reviewers’ constructive comments:
- We acknowledge the Reviewer’s point that quantitative real-time PCR (qRT-PCR) provides greater precision compared to fluorescence-based methods in the evaluation of inflammation markers. Nonetheless, RNA isolation was not feasible in the principal segment of our experiment unless cell sorting is employed to separate distinct subpopulations. Notably, the mean sample values measured by qRT-PCR fundamentally differ from the fluorescence measurements obtained in individual cells via flow cytometry, since the latter approach preserves cell individuality and enables selective gating as per experimental objectives.Also our core experimental design focused on characterizing the distribution of two interrelated cellular features while maintaining single-cell resolution. Stimulation trials were conducted using lipopolysaccharide (LPS) to induce inflammation and tert-Butyl hydroperoxide (TBH) to elicit oxidative stress. Across all ECA1-4 groups, a comparable increase in IL-6 expression was observed, although responses to other markers were less consistent. These results indicate the presence of moderate inflammation, and more expressed oxidative stress, likely affected by the predefined experimental settings. Our other dataset not presented here comprises similar recordings from human monocytes, which exhibit pronounced inflammatory responses when exposed to ECA. In fact one significant limitation of our study is the lack of standardized cigarette data, which resulted from both time constraints during the study and institutional inertia that prevented timely completion. Future work incorporating a third inflammation marker may offer additional validation for the observed increase in this feature within the gated cell subpopulations
- NF-kB is a pleiotropic transcription factor that regulates inflammation by promoting the expression of pro-inflammatory genes such as cytokines, chemokines, and adhesion molecules. We aggree with the Reviewer that it should not be viewed or investigated solely as an on/off switch since its activity is modulated by multiple mechanisms, including various post-translational modifications, subcellular compartmentalization, and interactions with different cofactors or corepressors. Initially, we aimed to assess both phosphorylated and naïve NF-kB, expressing their values as a ratio on the y-axis of binary scatterplots alongside DCFDA measurements. Unfortunately, this ratio proved highly variable, primarily due to inconsistent signals from our monoclonal phospho-NF-kB p65 (Ser536) (93H1) antibody. Consequently, we decided to exclude this parameter from our binary analysis, especially since IL-6 measurements remained stable, simple and easily reproducible. Profilling NF-kB in context of transcriptional activation is certainly better but much more experiments are needed to use it. Instead, the intracellular content of NF-kB was simply recorded in a Table as an inflammatory mediator, without additional commentary.

Reviewer 2 Report
Comments and Suggestions for Authors
Although the authors state in responses that “The novelty and significance of our study have been further emphasized in the Introduction and Disscussion” there are no additions on this issue in the Introduction nor Discussion section, especially in aims of the study that emphasize the novelty of this study compared to previously published.
Next, the authors state in responses” The term “prospective diagnostic algorithm” has been removed to avoid overstatement. And yet, in the aims of the study in the Introduction section still stands “possible diagnostic algorithm of subclinical inflammation”. The same is in Discussion section that starts with” This study aimed to assess the proinflammatory and prooxidative effects of widely used EC aerosols in cultured human epithelial cells, compared with conventional CS, and to develop a diagnostic framework for identifying the most deleterious EC formulations and defining cell subsets for prospective diagnostic algorithms”. I must emphasize again that this can not be regarded as “diagnostic framework” nor “diagnostic algorithms”.
Method described 2.1.2 Intracellular IL-6, TNF-α and NF-κB : Add specifications for the used antibodies
In the Methodology section statement “We addressed the issue by single and double staining the cells using DCFDA and epitope-specific rabbit monoclonal antibodies conjugated to peridinin chlorophyll protein-cyanine5.5 (PC5.5), from Cell Signalling (Danvers, MA, USA), as described earlier” needs reference. Further, add specifications for the used antibodies and dilution ratio.
Also, provide in the Introduction description of this approach, why did you evaluate Binary Fluorescence Scatterplots of IL-6 vs Oxidative Stress, few words about the link of IL6 with increase of oxidative stress in smoke induced damage.
In Figure 1 there are no letters A,B and C in the images.
Section 3.2.1. Intracellular oxidative stress. Instead of “CS elevated DDCF fluorescence” correct to “CS elevated DCFD fluorescence”
In what units is the level of oxidative stress, how did you calculate it?
Description of results is problematic. Example: title “Intracellular TNF-a”, is not informative enough.Also, description starting “TNF-a was elevated” does not indicate what was exactly measured, ratio of TNF-a positive cells or something else. The same issue stands for Intracellular NF-kB and Intracellular IL-6 results.
Please elaborate on the rationale and provide supporting references for interpreting the double staining results in terms of zonal cell distribution (e.g., Zone I interpreted as high inflammation, low oxidative stress). The current interpretation seems overly generalized and lacks sufficient justification.
The Results and Discussion sections are generally unclear, offering vague interpretations that do not reflect established scientific facts. Example: “Oxidative stress provides a broad signal compared to the narrow molecular signal of IL-6.”
A thorough revision is required to ensure proper interpretation of the results and their contextualization through comparison with existing literature, thereby clarifying the significance of the findings.
Author Response
We gratefully acknowledge the reviewers’ constructive comments, which have substantially improved the quality and clarity of our manuscript. In response, we have implemented comprehensive revisions, in line with the reviewers’ recommendations. The revisions are summarized below:
- Novelty of the study:
To the best of our knowledge, this is the first study demonstrating significant differences in the biological effects of “watermelon” and “strawberry-flavored” electronic cigarettes. Unlike conventional cigarettes (CS), electronic cigarettes (EC) do not induce acute toxicity but rather subtle inflammation and oxidative stress that may promote biochemical alterations, thereby adversely affecting consumer health. It is therefore conceivable that EC exposure contributes to subclinical inflammatory states not routinely detected in current clinical laboratory practice. For this reason, we initially introduced the concept of “diagnostic algorithms” aimed at monitoring such subtle biochemical changes. This manuscript should be regarded as preliminary and primarily “technical” in scope. The intention was not to explore molecular mechanisms but to utilize selected biomolecular markers for analytical purposes. This approach is important, as EC-related biomarkers are not part of standard laboratory diagnostics. The situation may be analogous to ethanol research, which, despite over a century of study, revealed that although ethanol is not toxic to cultured cells, it significantly affects cellular redox balance, but general mechanism in not known.
A dual-parameter fluorescence scatterplot analysis may provide a novel perspective in clinical diagnostics, offering new tools even if their clinical relevance remains to be fully established.
Additional explanatory details on this subject have been added to the Introduction and Discussion sections, emphasizing the novelty of the study.
- Diagnostic framework:
Embedded within preclinical and in vitro research, our objective was to highlight new analytical aspects and introduce innovative tools applicable to routine investigations. The term “prospective diagnostic algorithm” was removed from the Conclusions and replaced with “potential diagnostic algorithm” in the Introduction, reflecting our belief that the described method represents a simple but sensitive analytical tool suitable for both research applications and future diagnostic use. While integrating numerical and graphical analyses is highly promising and offers several advantages, all direct references to “diagnostic frameworks” and “diagnostic algorithms” have been removed, given the preliminary nature of the findings.
- Antibody specification:
Detailed information regarding antibodies used for IL-6, TNF-α, and NF-κB analyses has been added to section 2.1.2.
- Double staining:
Information on the origin and dilution of the IL-6 antibody applied in the double staining with DCFDA and IL-6 PC5.5 antibody has been included.
- IL-6 and oxidative stress relationship:
Over 18,000 studies have investigated IL-6 and its relationship with oxidative stress, demonstrating that IL-6 can either activate or suppress oxidative stress depending on the biological context. Correlations between IL-6 expression and reactive oxygen species levels have also been observed in studies involving PM2.5 exposure. Although we did not explore mechanistic interactions between IL-6 and oxidative stress, we used both markers diagnostically, and the approach proved effective. Our efforts to incorporate additional molecular markers were unsuccessful, as their regulatory mechanisms are considerably more complex and fall outside our area of expertise. A concise explanation and relevant reference regarding the IL-6/oxidative stress link have been added to the Discussion section.
- Figure labeling:
Figure 1 now includes panels A, B, and C for clarity.
- Typographical correction:
The term “DDCF” has been corrected to “DCF” (Results section 3.2.2.1).
- Units of oxidative stress:
Oxidative stress was expressed in relative units. The results represent mean fluorescence intensity (± SD) for each sample, derived from histograms commonly used in single-parameter flow cytometric analyses. Histograms displaying DCF signal intensity (x-axis) versus cell count (y-axis) were used to represent data. A rightward shift of the peak indicates higher fluorescence intensity, corresponding to increased oxidative stress. A positive control in this assay was conducted using tert-butyl hydroperoxide.
- Intracellular cytokine quantification:
For intracellular TNF-α, IL-6, and NF-κB measurements, the results represent mean epitope expression in analyzed cell population, typically based on 10,000 cells. This information has been added to the Materials and Methods section.
- Interpretation of results:
Flow cytometry analysis is complex (O'Neill K et al. Flow cytometry bioinformatics. PLoS Comput Biol. 2013,9,e1003365). Gating-the process of selecting subpopulations within the total cell population in flow cytometry is inherently relative. Quantified fractions obtained from gating can be analyzed to compare different defined cell populations. Statistical evaluation of these data supports robust interpretation, allowing biologically meaningful insights to emerge from distinct cell subsets. For example, Zone I may represent cells with high inflammation but low oxidative stress, whereas Zone IV may exhibit the opposite profile. Within the same dataset, one cell in Zone I (red arrows) showed TNF-α = 200 relative units and oxidative stress = 4000, while a cell in Zone IV (black arrows) displayed TNF-α = 180 and oxidative stress = 8000. We have demonstrated that this relative gating framework enables effective comparisons among different subpopulations. Corresponding figure is attached in pdf file.
- Clarification of reviewer’s comment:
In response to the reviewer’s remark regarding the interpretation “Oxidative stress provides a broad signal compared to the narrow molecular signal of IL-6,” this sentence was originally included in the Limitations of the Study section to highlight the contrast between the strong, cell-wide “inorganic” signal from DCF and the subtler, “organic” signal derived from IL-6. We have no idea about reviewer’s interpretation of the term “established scientific fact’ in this context, but this distinction has now been clarified in the revised version od the MS.

Round 3
Reviewer 2 Report
Comments and Suggestions for Authors
The explanations from responses no 1 and 2 to previous issues should have been incorporated in the text of the manuscript, to clarify to the readers about the novelty and diagnostic framework. Following sentences from responses should be included in manuscript: “Embedded within preclinical and in vitro research, our objective was to highlight new analytical aspects and introduce innovative tools applicable to routine investigation.”
“To the best of our knowledge, this is the first study to reveal significant differences in the biological effects of “watermelon” versus “strawberry-flavored” electronic cigarettes (ECs). Unlike conventional cigarettes (CS), which are known to cause acute toxicity, ECs appear to induce more subtle forms of inflammation and oxidative stress. These changes may lead to biochemical alterations that negatively impact consumer health, potentially contributing to subclinical inflammatory states that are not routinely detected by current clinical laboratory methods. To address this gap, we introduced the concept of “diagnostic algorithms” designed to monitor these nuanced biochemical shifts. This approach is particularly relevant given that EC-related biomarkers are not yet part of standard diagnostic protocols.”
The interpretation of flow cytometry results should be revised. The current description of cell cycle distribution is overly generalized. For example, “Cells were classified based on their relative distribution in the individual cell cycle phases: damaged, subdiploid G0/G1 zone (early G0/G1 cells) – cytotoxicity, diploid zone (G0/G1 - before DNA synthesis/resting). “ That explanation of damaged cells is somewhat vague and could benefit from clarification and more precise terminology. E.G. Subdiploid DNA content typically indicates apoptotic cells. On the other hand, DNA damage activates checkpoints (e.g., G1/S, G2/M) that halt progression to allow repair, and indicates presence of cell cycle arrest due to DNA damage. Thus, subdiploid DNA content typically reflects apoptosis, while DNA damage leads to cell cycle arrest at specific checkpoints (e.g., G1/S, G2/M) without necessarily reducing DNA content. Please clarify this distinction and adjust the interpretation of your findings accordingly.
Since you are presenting these findings as new technical approach in risk assessment, the validity of the FACS technique should be confirmed by comparison with immunocytochemical staining, at least for ECA4 that showed highest change.
Author Response
We gratefully acknowledge the Reviewers’ constructive comments. The revisions are summarized below:
- Sentences from responses 1 and 2 have been integrated into the manuscript to better inform readers about the novelty and overall diagnostic framework addressed in the study .
- The initial description of cell cycle distribution was overly broad, as this measurement was used exclusively to confirm specific type of exposure. Our group has been analyzing the cell cycle in cancer models since 1995. In cell-based experiments, establishing the exposure model involves understanding that overexposure generally causes extensive cell death, while subtle changes may not be evident. For EC, cytotoxicity tends to be low, whereas CS can induce considerable toxicity. The interpretation of flow cytometry results has now been revised to clarify that the subdiploid cell fraction indicates DNA damage or apoptotic cell death, while accumulation of cells in G1/S and G2/M phases signifies cell cycle arrest .
- The validity of the FACS technique in EC studies has been confirmed by multiple authors. Flow cytometry is more suitable for biomonitoring toxic effects in cells grown in suspension, while immunocytochemistry is less appropriate for tissue samples. Additionally, toxicity of ECA4 in cultured cells has been documented by others, and three relevant references have now been included into the MS.

Round 4
Reviewer 2 Report
Comments and Suggestions for Authors
The author provided non satisfactory response no 3 "Flow cytometry is more suitable for biomonitoring toxic effects in cells grown in suspension, while immunocytochemistry is less appropriate for tissue samples. ", since you used adherent cells and not suspension of cells, you could easily provide confirmation by immunocytochemistry and visualize signals by fluorescent microscopy. However, since the authors insist of not performing additional experiments, I would advise the editor to consider this point.
Other corrections are adequate.